Change characteristics and influencing factors of grassland degradation in adjacent areas of the Qinghai–Tibet Plateau and suggestions for grassland restoration

Lin Gang 1 2
Hua Limin 1
Shen Yanze 1
Zhao Yajiao 1 zhaoyj@gsau.edu.cn
1 College of Pratacultural Science, Gansu Agricultural University , Lanzhou, No. 1 Yingmen Village , China
2 Department of Gansu Natural Resources Planning and Research Institute , Lanzhou , China
Orlov Yuriy
Electronic publication date: 2023 Sep 12
Publication date: 2023
Volume: 11
Electronic Location ID: e16084
Received 2022 Dec 19; Accepted 2023 Aug 21
Copyright: © 2023 Lin et al.
Copyright year: 2023
Copyright holder: Lin et al.
License: This is an open access article distributed under the terms of the Creative Commons Attribution License, which permits unrestricted use, distribution, reproduction and adaptation in any medium and for any purpose provided that it is properly attributed. For attribution, the original author(s), title, publication source (PeerJ) and either DOI or URL of the article must be cited.
License URL: https://creativecommons.org/licenses/by/4.0/

Keywords: Qinghai–Tibet plateau, Grassland, Climate change, Socioeconomic factor, Ecological education

Funding: National Natural Science Foundation of China 32160338 Doctoral Research Start-up Fund GAU-KYQD-2020-18 This work was supported by the National Natural Science Foundation of China (32160338) and the Doctoral Research Start-up Fund (GAU-KYQD-2020-18). The funders had no role in study design, data collection and analysis, decision to publish, or preparation of the manuscript.

==============================
Natural grasslands are being progressively degraded around the world due to climate change and socioeconomic factors. Most of the drivers, processes, and consequences of grassland degradation are studied separately, and it is not yet clear whether the change characteristics and influence factors of adjacent areas of grassland are identical. We analyzed changes in grassland area and quality, and the influences of climate changes and socioeconomic factors from 1980–2018 in Maqu County, Xiahe County and Luqu County on the eastern Qinghai-Tibet Plateau (QTP). We found that areas with high and medium coverage grassland in Maqu County and Luqu County decreased continuously with time, while low coverage grassland areas increased in three counties. In Xiahe County, the medium coverage grassland area reduced with time (except for 2010), while the high and low coverage grassland areas increased. The actual net primary productivity of the three counties showed a downward trend. In Maqu County, the total grassland area had an extremely significant positive correlation with number of livestock going to market, commodity rate, gross domestic product (GDP), primary industry, tertiary industry, household density, and levels of junior middle school education and university education in the area. In Luqu County, the total grassland area high coverage grassland area were significantly negatively correlated with total number of livestock, secondary industry, levels of primary school education, and temperature. Ecological education was positively correlated with high coverage grassland, and negatively correlated with low coverage grassland in all three areas. The results of this study suggest that the best ways to restore the area and quality of grasslands in these areas would be to reduce the local cultivated land area and slow down the development of the primary and tertiary industries in Maqu County, and to control industry development and the total number of livestock in Luqu County. This study also suggests that improving education level and strengthening the level of ecological education are conducive to the restoration of grasslands.

Introduction

Grassland ecosystems are one of the most widely distributed vegetation types, accounting for about one-fifth of the world’s surface area (Zhang et al., 2015). They play a vital role in maintaining biochemical cycles, regulating climate, preventing desertification, protecting biodiversity, conserving water, and supporting animal husbandry and food production (Fayiah, Dong & Khomera, 2022). However, it has been estimated that nearly half of the world’s grassland ecosystems are being degraded, with around 5% experiencing strong to extreme levels of degradation (Gang et al., 2014)—especially on the Qinghai-Tibet Plateau (QTP; Fassnacht, Li & Fritz, 2015). Numerous studies have been carried out to analyze grassland degradation worldwide (Gao et al., 2013). The QTP has an area of approximately 2.5 million km2, which is nearly 25% of China’s total area. It also has a high-altitude and a long history of grazing (Li & Song, 2021). As one of the main pastoral regions of China, 60% of the QTP area is composed of grassland (Shen et al., 2019), which serves as an important terrestrial ecological barrier in China (Ren, Lü & Fu, 2016). The rational utilization of grassland resources and sustainable development of animal husbandry play an important role in maintaining national stability, border security and grassland culture inheritance on the QTP. Therefore, research on the QTP grassland ecosystem is essential, especially with the observed rise in pest and rodent outbreaks, decreases in biomass, biodiversity, and vegetation cover, as well as increases in soil nutrition, soil erosion, sandstorms, greenhouse gases and ecosystem services (Fedrigo et al., 2017; Hu, Zhang & Wang, 2017; Hopping et al., 2018). Given its relevance, QTP grassland degradation has become an emerging topic in the fields of environmental protection and grassland management.

Since ecosystem vulnerability is generally caused by both natural and human activities, the driving factors behind grassland degradation can be divided into two categories: climate changes and socioeconomic factors (Kang et al., 2018; Chuvieco et al., 2014). However, the relative contributions of grassland degradation remain poorly understood. The grassland ecosystem of the QTP is extremely sensitive to climate changes (Jin, Jin & Mao, 2019). Scientists generally believe that climate changes are driving grassland degradation on the QTP (Wang et al., 2020a), and that the main climate changes are temperature and precipitation (Che et al., 2018). However, the influencing factors of climate in different areas of grassland are spatially heterogeneous. For example, Li et al. (2019) found that the southern alpine grasslands showed a strong response to temperature, whereas the northeast alpine grassland was sensitive to precipitation, and the intermediate alpine grassland was mainly affected by radiation and temperature. Therefore, the impacts of climate change on grassland degradation are different in different areas. Even though certain areas of the QTP have been well studied in terms of grassland degradation, most studies have not considered anthropogenic factors (Xia et al., 2021). Livestock grazing is one of the most important factors affecting grassland degradation in the QTP, as it reduces grassland production and vegetation cover, leading to a reduction in soil quality (Wang et al., 2020b). However, some studies suggest that overgrazing is just a symptom, and that changes in grassland management policies and human and social development are the main reasons for grassland degradation. In the 1990s, assessments of ecosystem vulnerability started considering socioeconomic factors (Qang, Yan & da Fang, 1999), including human activities and development and social and economic developments. This is key because when the disturbance intensity of socioeconomic factors exceeds the carrying capacity of the ecosystem, the ecological environment is degraded, resulting in increased grassland degradation (Guo et al., 2016). However, studies on the impacts of socioeconomic factors on grassland degradation—especially the impacts of grassland management policies—are lacking. There is a strong need for studies on the effects of socioeconomic factors on grassland degradation in the QTP region.

Current knowledge on grassland degradation and its influencing factors are mostly based on large-scale studies, with relatively few small-scale studies (Chen et al., 2014; Gao et al., 2013). However, grassland change characteristics and influencing factors differ based on unique local topography, social development and grazing level, even between adjacent grasslands in the same pastoral regions. For example, different policies and economic development orientations in different regions lead to different levels of animal husbandry and different degrees of grassland degradation or restoration (Fayiah et al., 2020). The number of local households, the ratio of men and women, and the distribution of labor also affect grazing and grassland quality, and education levels determine people’s understanding of grassland protection, but the majority of studies about the influencing factors of grassland degradation or restoration do not consider these types of influences. Therefore, local education levels were also considered when discussing the influencing factors of grassland degradation in this study.

The grasslands of Maqu County (MQ), Xiahe County (XH) and Luqu County (LQ) are connected, but the climate conditions are slightly different and differences in regional economic development are significant. In this study we sought to explore whether the level of grassland degradation and most significant influencing factors were similar in these adjacent grasslands that have similar climate conditions but different social developments. Total grassland area, ecosystem vulnerability and productivity were studied in these three adjacent areas (MQ, XH and LQ) in the east of the QTP and the factors affecting these changes were analyzed, including both climate factors (temperature, precipitation and water use efficiency) and socioeconomic factors (livestock, population, household, GDP and education; Fig. 1). Suggestions for improvements were then put forward for each local government based on the results of the analysis (Fig. 2). This study is also a useful reference for the maintenance of grassland ecosystem stability and helps lay a foundation for the future formation of rational grassland management systems.

Figure 1 Impacts of climate changes and socioeconomic factors on grassland degradation.

Figure 2 Scientific assumptions and research ideas.

Data and methods

Study region

The study area included three adjacent grasslands—MQ, XH and LQ—located in Gannan Tibetan Autonomous Prefecture in southwestern Gansu Province, China (Fig. 3). All three are located in the transition area between the northeastern edge of the QTP and the western part of the Loess Plateau (100°45′–103°25′E, 33°06′–35°34′N). The areas are composed of vast grasslands and are typical pastoral areas of the QTP. The average altitude in the study areas is 3,600–3,800 m, with the highest average altitude in MQ, followed by XH and then LQ. Changes in the average annual temperatures are consistent with the average altitudes: MQ has the largest change (5.07 °C), followed by XH (2.93 °C) and then LQ (2.76 °C). Average annual precipitation is correlated with the latitude of each region with MQ having the highest (602.53 mm), followed by LQ (592.74 mm) and then XH (536.78 mm). XH has the highest GDP of the three areas, followed by MQ and then LQ, while MQ has the largest GDP per capita, followed by LQ and then XH.

Figure 3 Geographical location of the study area.

The areas of MQ and LQ are 10,678 hm2 and 4,817 km2, respectively. The area of XH changed in 1996, when the Ministry of Civil Affairs approved the establishment of a Hezuo city, leading to the separation of Hezuo town and seven townships from XH. Following this adjustment, the total area of XH became 6,959 km2. The grassland areas in MQ and LQ comprise a large proportion of the total area (73.00% and 73.87%, respectively), while only 63.22% of XH is grassland area. However, the proportion of cultivated land to the total area, and the proportion of forested land to the total area, are opposite to the trend in the proportion of grassland to the total area, with XH leading followed by LQ and then MQ. The proportions of cultivated land to the total area in MQ, LQ and XH are 0.01%, 0.51% and 4.42%, respectively, and the proportion of forested land to the total area is 8.00%, 22.01% and 29.95%, respectively. There are seven types of grassland in MQ (e.g., alpine meadow, marsh meadow, etc.), with alpine meadow having the largest area and distribution. Alpine meadow and marsh meadow constitute the main grassland types in XH. There are also seven grassland types in LQ, including alpine meadow, marsh meadow and shrub meadow, with alpine meadow the main type.

Data collection

The data used in this study included remote sensing, land type, topographic, meteorological, and socioeconomic data from 1980 to 2018. Specifically, this study used visual interpretations of 30-m-resolution Landsat image data to determine area sizes of land types and ecosystem vulnerability levels of grasslands in the study region. The Landsat Thematic Mapper images were in vector format with a scale of 1:100,000, and had an average interpretation accuracy of 92.9% (Liu et al., 2003). The data were resampled to a spatial resolution of 500 m. The remote sensing data included the Normalized Difference Vegetation Index (NDVI) obtained from a Moderate Resolution Imaging Spectroradiometer (MODIS; Lei, Zeng & Zhang, 2012). All data were processed using ARCGIS 10.2 software (Li & Song, 2021). First, all data were projected into the same coordinate system (WGS 1984 UTM 45N) and then divided using spatial boundaries according to study area. Finally, the spatial resolution of the data was unified to 1 km by bilinear interpolation. The NDVI dataset comprises monthly data with 12 periods per year, and the annual NDVI was generated by selecting the annual maximum.

The meteorological data were derived from China’s Meteorological Data Sharing Service System (Beijing, China), which included the average annual temperature, average annual precipitation, and total solar radiation in the study area.

The number of livestock taken to market, total number of livestock and the commodity rate were obtained from the Third, Fourth, Fifth, Sixth, Seventh National Census in China.

Data on GDP, primary industry, secondary industry, tertiary industry, population, number of households, ratio of men to women, number of villages/towns, and education level were obtained from the “Gannan Yearbook.”

Calculation of NPP

Net primary productivity (NPP) is a fundamental indicator of vegetation productivity that can reflect vegetation dynamics and the status of ecological processes. To identify the impacts of climate change and human activities on grassland change, three kinds of NPP were defined, as outlined by Li et al. (2016). Actual NPP (ANPP) indicated the actual situation in which grassland productivity was affected by both climate and human activities. It was calculated using the Carnegie-Ames-Stanford Approach (CASA) model, which is a light-use efficiency model that uses remote sensing data, meteorological data, and vegetation types as the input parameters (Potter et al., 1993). Potential NPP (PNPP) indicated the hypothetical situation in which grassland productivity was affected only by climate and was calculated using the Thornthwaite Memorial model (Lieth & Box, 1972). Human-induced NPP (HNPP) indicated the hypothetical situation in which grassland productivity was affected only by human activities and was calculated based on the methods outlined by Li et al. (2016). Detailed calculations are described in the Supplemental Materials.

Calculation of RUE

The RUE (g·m−2·mm−1) calculation is based on a simple regression:

RUE=ANPPP,

where “P” is the annual average precipitation (mm).

Abrupt change analysis

To check for changes in grassland area and NPP trends from 1980–2018, the non-parametric Mann-Kendall (M-K) test (Kendall, 1955; Mann, 1945) was adopted for change-point detection.

Results

Spatiotemporal variations in land types and grassland NPP

From 1980 to 2018, the cultivated land areas in MQ, XH and LQ increased with time (Table 1 and Fig. 4), with values in 2018 that were 2.31, 1.23 and 2.05 times what they were in 1980, respectively. The forested areas in MQ and LQ did not change significantly over the 48 years, while the forested area in XH significantly degraded (a reduction of 3.82% compared with 1980). The areas of urban–rural residential land also expanded in these counties over time, with MQ having the largest change, followed by LQ and then XH; the values in 2018 were 3.51, 2.49 and 1.38 times what they were in 1980, respectively. The proportions of grassland area in MQ and XH were basically stable from 1980–2018, maintaining at 73% and 63%, respectively. In LQ, the proportion of grassland was 72% in different periods, except in 2010 when it was 71%. The total grassland in MQ increased with time, by a total of 41.56 km2 in 2018 compared with 1980. The grassland area in XH changed slightly in different years. However, the grassland area in LQ decreased continually with time. By 2018, the grassland area in LQ had decreased by a total of 42.23 km2 compared with 1980. The turning point year for grassland area in MQ occurred in 2008, and in 2000 in LQ (Fig. 5).

Table 1 Areas of different land cover classifications in Maqu, Xiahe and Luqu (hm2) counties.

Region	Year	Cultivated land	Forest	Grassland	Water area	Urban–rural residential land	Unutilized land	
Maqu County	1980	0.69	844.07	7,796.12	186.14	4.69	1,870.22	
1990	0.70	862.11	7,793.97	172.82	4.69	1,867.65	
2000	2.10	855.10	7,805.95	163.28	6.07	1,869.45	
2010	1.58	855.62	7,813.45	199.96	12.56	1,818.79	
2018	1.59	856.71	7,837.28	175.88	16.48	1,790.38	
Xiahe County	1980	268.49	2,137.58	4,399.44	8.22	28.26	122.03	
1990	269.06	2,137.87	4,398.58	8.24	28.27	122.02	
2000	338.16	2,045.11	4,421.13	8.30	30.50	120.98	
2010	332.54	2,050.28	4,389.81	8.20	35.74	147.58	
2018	330.11	2,055.93	4,402.46	8.13	39.07	123.75	
Luqu County	1980	14.00	1,061.17	3,489.63	21.10	6.23	233.58	
1990	14.00	1,060.42	3,489.98	21.10	6.23	233.97	
2000	27.57	1,047.37	3,489.51	18.73	7.69	234.84	
2010	38.11	1,069.01	3,420.08	35.02	12.34	251.23	
2018	28.69	1,070.00	3,446.40	34.67	15.54	221.45	

Figure 4 Areas of different land cover classifications in Maqu, Xiahe and Luqu counties (hm2).

The pictures from top to bottom depict Maqu County, Xiahe County and Luqu County; from left to right, the pictures represent the years 1980, 1990, 2000, 2010 and 2018.

Figure 5 Change−point detection of grassland area in Maqu County, Xiahe County and Luqu County from 1980−2018.

The pictures from top to bottom depict the total grassland area, the high coverage grassland area, the medium coverage grassland area and the low coverage grassland area.

Areas of high and medium coverage grassland in MQ and LQ decreased continuously with time, while the areas with low coverage grassland increased in these counties (Table 2 and Fig. 6). In 2000, low coverage grassland areas totaled 1,195.26 km2, but in 2010 this had risen to 1,431.23 km2—an increase of 235.97 km2 in those 10 years. In XH, the total area of medium coverage grassland decreased with time (except in 2010), while the area of low coverage grassland increased. Areas of high coverage grassland increased in XH from 1990 to 2000, and then declined after 2000. Among the three counties, the high, medium and low grassland coverage areas as a proportion of the total grassland area in MQ were 24.63%, 58.17% and 17.21%, respectively; the proportions in XH were 58.15%, 40.11% and 1.74%, respectively, and were 80.53%, 19.47% and 0.003% in LQ, respectively. The growth rate of low coverage grassland areas was largest in MQ, followed by XH and then LQ. At a regional scale, the turning point year of high coverage grassland in XH occurred in 2008, and in 2000 in LQ (Fig. 5). In XH and LQ, 2000 was the turning point year of medium coverage grassland, and 2008 was the turning point year for low coverage grassland.

Table 2 Changes in ecosystem vulnerability levels of grasslands in Maqu, Xiahe and Luqu counties (hm2).

Year	Maqu County	Xiahe County	Luqu County	
High coverage grassland	Medium coverage grassland	Low coverage grassland	High coverage grassland	Medium coverage grassland	Low coverage grassland	High coverage grassland	Medium coverage grassland	Low coverage grassland	
1980	1,955.13 (25.08%)	4,643.53 (59.56%)	1,197.47 (15.36%)	2,524.10 (57.37%)	1,795.36 (40.81%)	79.99 (1.82%)	2,811.68 (80.57%)	677.96 (19.43%)	0.00 (0.00%)	
1990	1,951.08 (25.03%)	4,647.64 (59.63%)	1,195.26 (15.34%)	2,522.73 (57.35%)	1,796.04 (40.83%)	79.81 (1.81%)	2,811.56 (80.56%)	678.42 (19.44%)	0.00 (0.00%)	
2000	1,893.22 (24.25%)	4,481.51 (57.41%)	1,431.23 (18.34%)	2,589.12 (58.56%)	1,750.35 (39.59%)	81.66 (1.85%)	2,813.51 (80.63%)	676.01 (19.37%)	0.00 (0.00%)	
2010	1,898.10 (24.29%)	4,468.17 (57.19%)	1,447.19 (18.52%)	2,587.01 (58.93%)	1,745.36 (39.76%)	57.44 (1.31%)	2,752.18 (80.47%)	667.54 (19.52%)	0.36 (0.01%)	
2018	1,917.09 (24.46%)	4,472.18 (57.06%)	1,448.02 (18.48%)	2,577.58 (58.55%)	1,740.63 (39.54%)	84.25 (1.91%)	2,771.50 (80.42%)	674.68 (19.58%)	0.22 (0.01%)	

Figure 6 Changes in different ecosystem vulnerability levels of grassland in Maqu, Xiahe and Luqu counties.

The pictures from top to bottom depict Maqu County, Xiahe County and Luqu County; and from left to right, the pictures represent the years 1980, 1990, 2000.

ANPP, PNPP and HNPP changed differently in XH, MQ and LQ and in different years (Fig. 7). Overall, the ANPP of the three counties showed a downward trend, and the ANPP of XH declined the most (44.58% lower in 2015 than in 1986). The PNPP of MQ and XH showed an upward trend, while the change in the PNPP of LQ was flatter during different years. HNPP showed an increasing trend in MQ, a decreasing trend in LQ, and showed smaller changes in XH between years. There were several abrupt change points of grassland NPP (ANPP, PNPP and HNPP) between 1980 and 2017 (Fig. 8), and the turning point year of NPP differed by region.

Figure 7 Changes in ANPP, PNPP, HNPP in Maqu, Luqu and Xiahe counties (g C·m−2·yr−1).

Figure 8 Change−point detection of NPP in Maqu County, Xiahe County and Luqu County from 1980−2018.

The pictures from top to bottom depict ANPP, PNPP and HNPP.

Spatiotemporal variations in climate changes

From 1980 to 2018, the average annual temperature in XH, MQ and LQ all increased with time (Fig. 9) by 0.03 °C·yr−1, 0.025 °C·yr−1 and 0.013 °C·yr−1, respectively. The average annual temperature from 1980–1990 in XH, MQ and LQ was 2.27 °C, 5.21 °C and 3.2 °C, respectively, and 3.48 °C, 5.19 °C and 3.12 °C from 2010–2018, respectively. From 1980–2010, the average annual temperature of XH increased rapidly (0.48 °C·10 yr−1), but increased relatively slowly from 2010–2018 (0.25 °C·10 yr−1). In MQ and LQ, the average annual temperature mainly increased between 1990 and 2010 (0.22 °C·10 yr−1 and 0.73 °C·10 yr−1), but changed only slightly in other time periods.

Figure 9 Changes in average annual temperature in Maqu, Luqu and Xiahe counties.

The average annual precipitation increased in MQ and XH from 1980 to 2018 (Fig. 10). The average annual precipitation in XH increased by 17.32 mm between 1990–2000 compared to 1980–1990, and by 39.40 mm in 2010–2018 compared to 2000–2010, with relatively small changes during other time periods. In MQ, the average annual precipitation decreased by 52.7 mm in 1990–2000 compared to 1980–1990, and by 74.6 mm compared to 1990–2000. In LQ, the average annual precipitation increased by 40.5 mm in 2010–2000 compared to 1990–2000, by 24 mm in 1990–2000 compared to 1980–1990 and by 5.3 mm in 2018–2010 compared to 2000–2010.

Figure 10 Changes in precipitation in Maqu, Luqu and Xiahe counties.

The RUE increased in all three counties over time (Fig. 11). In the four time periods of 1980–1990, 1990–2000, 2000–2010 and 2010–2018, the RUE in MQ was 0.67, 0.73, 0.94 and 0.92 g·m−2·mm−1, respectively; 0.82, 083, 1.16 and 1.04 g·m−2·mm−1 in XH, respectively; and 0.77, 0.84, 0.98 and 1.01 g·m−2·mm−1 in LQ, respectively.

Figure 11 Changes in RUE in Maqu, Luqu and Xiahe counties.

Spatiotemporal variations in socioeconomic factors

This study found that the number of livestock taken to market and the commodity rates of livestock in the three counties both increased from 1980 to 2018 (Fig. 12). The number of livestock taken to market in XH, MQ and LQ increased rapidly from 2010 to 2018—by 90%, 75% and 97%, respectively, compared to 2000–2010. The total number of livestock also increased in XH, MQ and LQ from 1980 to 2008 and reached their peaks in 2008 (1,018,500; 1,060,000 and 729,900 heads, respectively). After 2008, the number of livestock slowly declined in all three counties.

Figure 12 Changes in livestock numbers in Maqu, Luqu and Xiahe counties.

In 1980, the GDP of MQ was 15 million yuan and the GDP of LQ was 11 million yuan. By 2018, the GDPs of MQ and LQ grew 105.73 and 92.36 times, to 1.586 billion yuan and 1.016 billion yuan, respectively (Fig. 13). GDP growth in XH was relatively slow from 1980 to 1991, growing from 28 million yuan to 89 million yuan, but increased from 1991 to 2018, growing from 89 million yuan to 1.660 billion yuan. The proportion of primary industry output in XH, MQ and LQ gradually decreased, while tertiary industry output gradually increased. The percentage of primary industry output in XH and LQ was lower than 50% by 2004, and in MQ it was lower than 50% in 1996. GDP per capita also increased continually in all three counties from 1980 to 2018, increasing rapidly in MQ after 1995, and in XH and LQ after 2005. By 2018, the GDP per capita in XH, MQ and LQ had increased by 72, 45 and 51 times compared to 1980, respectively.

Figure 13 Changes in GDP and industrial output in Maqu, Luqu and Xiahe counties.

From 1980 to 2018, the population, population density, number of households and household density in MQ and LQ showed an upward trend (Fig. 14). In 2018, compared to 1980, the population of MQ and LQ increased by 132.8% and 71.7%, respectively, and the number of households increased by 231.4% and 108.5%, respectively. The population of XH was impacted by the establishment of Hezuo city in 1996, which involved one town and seven counties originally belonging to XH being divided into the jurisdiction of Hezuo city, leading to a large population decrease in XH in 1997. However, the population and population density increased in XH from 1980 to 1997, and then continued to increase from its new (lower) level in 1997 to 2018. XH, MQ and LQ all showed higher numbers of men than women (Fig. 15). The urban populations of the three counties were significantly smaller than their rural populations in 1990, 2000 and 2010, although the urban and rural populations of MQ and LQ were similar in 2018.

Figure 14 Changes in population and household numbers in Maqu, Luqu and Xiahe counties.

Figure 15 Changes in population structure in Maqu, Luqu and Xiahe counties.

The proportion of the population that were educated increased with time in MQ and LQ (Fig. 16); whereas, in XH, this proportion increased from 2000 to 2010, and then decreased slightly after 2010. However, the proportion of the population with a college education increased in all three counties from 1980 to 2018.

Figure 16 Changes in education levels in Maqu, Luqu and Xiahe counties.

Correlation analysis

The high coverage grassland and cultivated land area had an extremely significant negative correlation in MQ, and an extremely significant positive correlation in XH (P < 0.01; Table 3). In LQ, total grassland area had an extremely significant negative correlation with low coverage grassland area, and an extremely significant positive correlation with high coverage grassland area (P < 0.01).

Table 3 Correlation analysis of land types.

		Cult.	Forest	Grass.	Water	U/R	Utilize.	H-G	M-G	
Maqu County	Forest	0.196								
Grass.	0.552	0.186							
Water	−0.241	−0.313	0.054						
U/R	0.458	0.221	0.955*	0.312					
Utilize.	−0.356	−0.232	−0.934*	−0.349	−0.993**				
H-G	−0.961**	−0.236	−0.505	0.013	−0.492	0.397			
M-G	−0.922*	−0.214	−0.763	−0.059	−0.745	0.667	0.942*		
L-G	0.922*	0.224	0.771	0.045	0.750	−0.672	−0.938*	−1.000**	
Xiahe County	Forest	−1.000**								
Grass.	0.315	−0.299							
Water	−0.096	0.100	0.585						
U/R	0.721	−0.723	−0.223	−0.755					
Utilize.	0.396	−0.414	−0.645	−0.248	0.452				
H-G	0.997**	−0.998**	0.289	−0.064	0.696	0.446			
M-G	−0.979**	0.979**	−0.182	0.298	−0.847	−0.429	-0.969**		
L-G	−0.263	0.282	0.627	0.061	−0.237	−0.973**	−0.325	0.264	
Luqu County	Forest	0.289								
Grass.	−0.844	−0.749							
Water	0.718	0.862	−0.954*						
U/R	0.735	0.681	−0.833	0.923*					
Utilize.	0.429	0.027	−0.395	0.124	−0.156				
H-G	−0.823	−0.779	0.998**	−0.970**	−0.854	−0.348			
M-G	−0.912*	−0.487	0.925*	−0.771	−0.626	−0.669	0.901*		
L-G	0.843	0.748	−1.000**	0.952*	0.827	0.404	−0.998**	−0.927*	
Note:

Asterisks (* and **) represent significance at the 0.05 and 0.01 (two-tailed) levels, respectively; Cult., Grass., U/R, H-G, M-G and L-G refer to cultivated land, grassland, urban–rural residential land, high coverage grassland, medium coverage grassland, and low coverage grassland, respectively.

Total grassland area in MQ had an extremely significant positive correlation with number of livestock going to market, commodity rate, GDP, primary industry proportion, tertiary industry proportion, household density and levels of junior middle school education and university education (P < 0.01; Fig. 17). The HNPP in MQ had an extremely significant negative correlation with high coverage grassland area (P < 0.01). In XH, total grassland area had an extremely significant negative correlation with primary school education (P < 0.01), and rural area was extremely positively correlated with medium coverage grassland area (P < 0.01). In LQ, the total grassland area had an extremely significant positive correlation with high coverage grassland area, and an extremely significant negative correlation with medium coverage grassland area, number of livestock, secondary industry proportion and primary school education (P < 0.01). Temperature and medium coverage grassland area had an extremely significant negative correlation in LQ (P < 0.01).

Figure 17 Correlation analysis of climate and socioeconomic parameters.

Cult., Grass., U/R, H-G, M-G, L-G, T, P, Marketing, Stock, Commodity, Primary, Secondary, Tertiary, Population-D, Household-D, P-E, J-E, H-E and U-E represent cultivated land, grassland, urban–rural residential land, high coverage grassland, medium coverage grassland, low coverage grassland, temperature, precipitation, number of livestock going to market, total number of livestock, commodity rate, primary industry growth, secondary industry growth, tertiary industry growth, population density, household density, and levels of primary school education, junior middle school education, high school education and university education, respectively.

Discussion

Change characteristics of grassland degradation in adjacent areas

Grassland degradation has become an emerging topic in the fields of grassland management and environmental protection (Manssour, 2011). The three areas in this study had different change characteristics in grassland area and productivity from 1980 to 2018. In these 38 years, the total grassland area in Maqu County increased by 0.53% (1.16 hm2), decreased by 1.24% (43.23 hm2) in Luqu County and did not change significantly in Xiahe County. This indicated that in the three adjacent areas of the Gannan Prefecture, the grassland areas were quite different and should be studied separately. In the study, grassland ecosystem vulnerability was divided into three levels, namely high, medium and low coverage grassland areas.

In Maqu County, the low coverage grassland area showed an increasing trend, while the high and medium coverage grassland areas showed decreasing trends. High coverage grassland area showed a significant positive correlation with medium coverage grassland area, and both showed a significant negative correlation with low coverage grassland area. These results indicate that the higher coverage grassland areas likely became lower coverage grassland areas in Maqu County, making the grassland ecosystem in Maqu County more vulnerable. This study also found that the high coverage grassland and medium coverage grassland areas decreased by a total of 209.39 hm2, while the low coverage grassland area increased by 250.55 hm2. This result indicates that other types of land, in addition to the high and medium coverage grassland areas, changed to low coverage grassland area.

In Xiahe County, high coverage grassland area increased by 53.78 hm2, medium coverage grassland area decreased by 54.763 hm2, while low coverage grassland area changed less (increased by 4.264 hm2). High coverage grassland area and medium coverage grassland area had an extremely significant negative correlation, indicating that part of the medium coverage grassland area was converted to high coverage grassland area in Xiahe County. In Luqu County, there were only high and medium coverage grassland areas from 1980–2000, with low coverage grassland first appearing after 2000. From 1980 to 2018, the high coverage grassland area decreased significantly, while the medium and low coverage grassland areas changed less. The total grassland degradation area was 43.23 hm2, while the high coverage grassland area was reduced by 41.18 hm2 in Luqu County. Total grassland area and high coverage grassland area had an extremely significant positive correlation, indicating that grassland degradation was mainly caused by the decrease in high coverage grassland area.

NPP reduction, as a core issue in grassland degradation, has been a main focus of research on ecosystem change (Wessels et al., 2007). NPP is affected by both climate changes and human activities (He, Richards & Zhao, 2016). Methods that compare the ANPP and PNPP of vegetation can determine the impact of humans on vegetation productivity (Li et al., 2016). ANPP represents the actual situation of vegetation productivity, influenced by both climate and human activities, while PNPP refers to the potential for plant growth in the absence of human disturbance and is only affected by climate changes (Xu et al., 2010). The difference between PNPP and ANPP is used to measure HNPP, or the impact of only human activity on vegetation productivity. In this study, the ANPP of the three grassland areas decreased gradually with time, falling the most in Maqu County, followed by Luqu County, while the ANPP of Xiahe County did not drop significantly. PNPP showed an upward trend with time in Maqu County and Xiahe County, indicating that the influence of climate changes on plant growth potential increased in Maqu County and Xiahe County over time. In our study, PNPP showed the same trend as temperature and precipitation, while ANPP showed an opposite trend due to the differences in geographical location and climate. Several scientists believe that human activity is the dominant factor affecting grassland degradation on the QTP (Pan et al., 2017). HNPP showed an increasing trend in Maqu County and Xiahe County, indicating that the impact of human activities on grassland vegetation productivity increased in these counties over time. In Luqu County, PNPP showed an increasing trend, while HNPP showed a decreasing trend, indicating that the impact of human activities was smaller than that of climate changes on grassland productivity in Maqu County. PNPP and HNPP measurements help quantitatively evaluate the driving forces behind grassland changes and help accurately identify the socioeconomic and climate change factors involved (Yuan, Yuan & Ren, 2021). In Maqu County from 1980 to 2018, the total grassland area increased, but the grassland quality was significantly reduced, and other land types were converted to low coverage grassland area. In Xiahe County, total grassland area increased, but the grassland quality was degraded slightly, and in Luqu County, total grassland area and quality were both degraded.

Influencing factors of grassland degradation in adjacent areas

The influencing factors of grassland degradation differed between Maqu, Xiahe and Luqu counties. Climate changes and human activities are not independent, with many coupled relationships existing between them (Xiong et al., 2019). Human activities have an important influence on land use changes on the QTP. In addition to animal husbandry and agriculture, changes in lifestyle brought on by urbanization, economic growth patterns, tourism and industrial activities are newer factors impacting grassland changes (Chuvieco et al., 2014). In Maqu County, many factors affected the total grassland area, for example, GDP, primary industry (mainly animal husbandry) and the tertiary industry all had an extremely significant impact on total grassland area. Increased GDP was followed by an increased demand for a higher quality of life. When the disturbance intensity of socioeconomic factors exceeds the carrying capacity of the ecosystem, the ecosystem becomes vulnerable and the grassland becomes degraded (Guo et al., 2016). Overgrazing is also one of the main causes of grassland degradation on the QTP (Liu et al., 2018). In this study, temperature and precipitation also impacted the total grassland area in Maqu County. Sufficient rainfall promoted the growth of grassland plants, leading to an increase in total grassland area (Eze, Palmer & Chapman, 2018). Chen et al. (2014) also found that rising temperature and precipitation promoted the expansion of grassland areas. When carrying out grassland restoration, it is important to consider both the characteristics of the grassland ecosystem itself as well as the impacts of climate changes (Jiang et al., 2016). This study also found that HNPP was negatively correlated with high and medium coverage grassland, and positively correlated with low coverage grassland. ANPP showed the opposite correlations as HNPP, indicating that socioeconomic factors, rather than climate changes, were the main cause of grassland changes on the QTP (Zhang et al., 2015). An increase in cultivated land means increased grassland, woodland and wetland damage. In this study, cultivated land had a negative impact on high and medium coverage grassland, and a positive impact on low coverage grassland, showing that cultivated land impacted grassland degradation. Chen et al. (2015a) also found that human activities, such as excessive grass reclamation, greatly impact the ecological environment. Future ecosystem protection should not ignore human interference, and sustainable human activity is an important factor in ecological restoration. An interesting phenomenon was also found in this study: the higher the education level of the local residents, the more favorable the increase of total grassland area and high/medium coverage grassland area. Improvements in education level may mean that people have a clearer understanding of ecological damage, grassland degradation and other hazards, and may improve people’s rationality of grassland use, thus reducing grassland degradation. In Maqu County, other types of land, in addition to both medium and high coverage grassland areas, were converted to low coverage grassland area. This conversion of other lands might be because social development and climate change destroyed the existing environment, transforming them into low coverage grassland areas. For example, excessive water use reduces total water area, with some of these areas becoming grasslands and deserts (the water area decreased by 6% during the 38 years). The total grassland area showed a significant negative correlation with total unutilized land, which decreased by 79.84 hm2 from 1980 to 2018. Likely some of the unutilized land was converted into grassland. Rising temperatures (0.96 °C from 1980–2018) may have also decreased snow coverage areas through melting, turning these areas into other types of land, including grassland. Similar to the results of other studies, the increased temperatures observed in this study promoted an increase in grassland area (Chen et al., 2015b). Sun et al. (2015) also observed significant changes in vegetation due to climate warming in the northwestern Loess Plateau.

In Xiahe County, high coverage grassland area had an extremely significant positive correlation with cultivated land, and an extremely significant negative correlation with forested and medium coverage grassland areas, mostly due to increases in cultivated land (61.624 hm2 total in the 38 years), leading to forest reduction (81.54 hm2 during 38 years). The increase in the population and number of households has led to the continuous increase in arable land, and the growth of the economy and livestock numbers have also forced herds to develop more grasslands (Gao et al., 2016). Herdsmen also urgently need to expand the area of cultivated land for pasture to improve the pasture utilization rate. Diabate et al. (2018) showed that the utilization rate of artificially planted pasture was three to four times that of natural pasture.

In this study, unutilized land and low coverage grassland had an extremely significant positive correlation, indicating that a part of the unutilized land was likely converted to low coverage grassland area. Xiahe County launched “The National Project of Returning Farmland to Forest (Grass)” in 1997, which led to a certain level of grassland restoration by 2000. High coverage grassland area had an extremely significant negative correlation with both the livestock commodity rate and the ratio of males to females in the local population. This is likely because a higher male to female ratio increased the demand for livestock products, so the livestock commodity rate also increased, forcing a higher grassland carrying capacity, thus reducing the grassland quality. Notably, in this study the male to female ratio had an extremely significant effect on high coverage grassland area, while population had no significant effect on high coverage grassland area, and population was not closely related to grassland area or quality. This can likely be explained by the administrative divisions in Xiahe County in 1997 (Hezuo city was established in 2000). This study also found that improving education levels could effectively promote grassland area increases, and improve the conversion of medium and low coverage grassland areas to high coverage grassland. This is because improving education level increased people’s protection of grasslands, improving grassland area and productivity.

In Luqu County, the total grassland area and high coverage grassland area decreased, while the medium and low coverage grassland areas only changed slightly. Water area had a significant negative correlation with total grassland and high coverage grassland, indicating that the total grassland and high coverage grassland areas might be caused by the increased water area that occupied the original high coverage grassland area. Also, as cities invade grasslands, their impervious surfaces change the reflectance and groundwater flow, breaking the original grassland landscape into fragments, causing high coverage grassland habitat loss. This study also found that the number of livestock had a significant impact on grassland area and quality, and was the main cause of the grassland degradation in Luqu County. To meet population demand and living standard growth, the number of livestock and cultivated land area increased, putting pressure on the grassland’s fragile ecological environment (Lin et al., 2020). This study showed that the total number of livestock number of livestock taken to market increased significantly between 1980 and 2018, exacerbating the livestock carrying capacity of grasslands for livestock, leading to grassland degradation. Livestock numbers peaked in 2008 and 2009, inevitably reducing grassland area and its ratio to total area. This study also showed that the secondary industry had an extremely significant negative impact on grassland area and quality. Specifically, industry development accelerated grassland degradation in Luqu County due to the increased CO2 emissions caused by the growth of the secondary industry. The increased CO2 concentration led to increased temperatures, causing grassland degradation and a reduction in high coverage grassland area. Gong et al. (2017) also found that secondary industry growth accelerated grassland degradation. Urbanization increases with secondary industry growth as more people are employed in the secondary industry, leading to increased CO2 emissions and rising temperatures (Yuan, Yuan & Ren, 2021). The impact of temperature on grassland degradation in Luqu County was more obvious than in Maqu County and Xiahe County. The increased temperature significantly reduced the total grassland and medium coverage grassland areas, but increased the low coverage grassland area. Some scientists believe that drought exacerbated grassland degradation on the QTP, because warming increases evaporation, prohibiting the growth of grassland vegetation (Chen et al., 2015b). Temperature increases also cause the permafrost to thaw, washing away the water in the soil, which is also not conducive to vegetation growth (You et al., 2017). Increased evaporation caused by warming could also significantly reduce the average soil moisture during the growing season (Fu et al., 2013). These findings indicate that rising temperatures increased the vulnerability of the grassland, and high coverage grassland areas gradually decreased, converting into medium and low coverage grassland areas. Luqu County has a lower altitude and a lower average temperature than Maqu County and Xiahe County, so the grassland ecosystem in Luqu County is more sensitive to temperature changes. The response of grassland to topography, climate change and human activities is a complex dynamic process. Geomorphic conditions impact temperatures and precipitation, leading to spatial differences in the grasslands of different areas (Zoungrana et al., 2018; Venkatesh et al., 2022). Education level also significantly affected grassland area and quality in all three counties included this study. This relationship between education level and grassland degradation has not been explored in previous studies.

Suggestions for grassland restoration in each of the three counties

The revegetation of degraded ecosystems is mainly attributed to environmental protection policies (Naeem et al., 2021). To effectively restore grasslands, different suggestions should be given to different areas based on the specific characteristics and influencing factors of grassland degradation in each area. In Maqu County, our recommendations are to reduce the cultivated land area, and slow down the primary industry (the total number of livestock, number of livestock taken to market and the commodity rate) and the third industry (such as Tourism). In Xiahe County, our recommendation is to reduce the commodity rate of livestock to improve grassland quality. In Luqu County, we suggest reducing the number of livestock and slow down secondary industry growth to improve the grassland area and quality. All three areas should improve the education level of the local population and improve awareness of ecological protection and sustainable use of grasslands.

Conclusions

In Maqu County, total grassland area increased, but the grassland quality decreased significantly from 1980–2018. The main factors behind these changes were increases in cultivated land, primary industry growth and tertiary industry growth. To restore grasslands in this area, the local cultivated land area should be reduced, the development of the primary and tertiary industries should be slowed, and ecological education should be improved. In Xiahe County, grassland degradation was not obvious, though improving the ecological education of the local people would still be conducive to grassland restoration. In Luqu County, grassland area and quality both declined between 1980 and 2018, primarily due to livestock production increases, secondary industry growth, and rising illiteracy rates and temperatures. Therefore, controlling the development of the local industry and the number of livestock, and improving the educational literacy of the local people can all help grassland restoration. This work provides a reference for the formulation of local policies on grassland protection and restoration.

Supplemental Information

Supplemental Information 1 Calculation of NPP.

Click here for additional data file.

Supplemental Information 2 Raw data.

Click here for additional data file.

Additional Information and Declarations

Competing Interests

Author Contributions

Data Availability

The authors declare that they have no competing interests.

Gang Lin conceived and designed the experiments, performed the experiments, analyzed the data, prepared figures and/or tables, authored or reviewed drafts of the article, and approved the final draft.

Limin Hua conceived and designed the experiments, performed the experiments, authored or reviewed drafts of the article, and approved the final draft.

Yanze Shen performed the experiments, prepared figures and/or tables, and approved the final draft.

Yajiao Zhao conceived and designed the experiments, performed the experiments, analyzed the data, prepared figures and/or tables, authored or reviewed drafts of the article, and approved the final draft.

The following information was supplied regarding data availability:

The raw measurements are available in the Supplemental Files.

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
