# Peer review of "Change characteristics and influencing factors of grassland degradation in adjacent areas of the Qinghai–Tibet Plateau and suggestions for grassland restoration"

_PeerJ, doi:10.7717/peerj.16084_

## Round 0.1 · original submission · Major Revisions

We have received 3 reviews on this work. There are some concerns demanding text updates, and changing the presentation. Please check the comments by reviewer #1. You are welcome to resubmit a revised version.

Reviewer 1 ·

Basic reporting

There were many areas in which the article needs to be improved. I have pointed out all the concerns below for your reference.

1. Firstly, the introduction top 2 paragraphs were very well framed and the remaining part of the introduction has to be totally reframed. The discussion is very well written and informative. The article is missing methods section and the implemented methodology is week.

Experimental design

2. Previous studies have already performed linear trend analysis or correlation analysis. What was the new/novelty that this current study is proposing to the scientific/research community?

3. In addition to trends and correlation which became quite common in literature, I strongly suggest authors to try incorporating models that can explain causation. Authors can even try to find the relative/absolute contributions of each climatic factor to vegetation phenology/NPP. Authors can try implementing PCMCI or geodetector or structural equation modeling or optimal fingerprint method to find these contributions. Adding either one of them will significantly increase the novelty of the article and might help for publishing.

4. Studies have shown that soil moisture, snow cover/depth, air temperature, soil types and livestock grazing will significantly impact vegetation (Both climate and socioeconomic drivers contribute to vegetation greening of the Loess Plateau; Optimal ranges of social-environmental drivers and their impacts on vegetation dynamics in Kazakhstan ; Spatiotemporal vegetation cover variations associated with climate change and ecological restoration in the Loess Plateau). Please refer to these studies and consider these factors in the current study or provide references and explain why the authors haven’t considered these factors in this study.

5. Are these trends mentioned related to linear trend analysis? It is standard these days to test non-linear trends in the study. I strongly suggest authors to implement Mann-Kendall and Sen's slope tests to find whether there is an increasing/decreasing trend and whether the slope is significant or not. Just linear trends are not acceptable and are not realistic in nature.

Validity of the findings

5. The entire manuscript is explaining the causation and authors performed correlation. Correlation cant explain causation. Hence it is mandatory to implement either one of the methods to explain causation and the article results and discussion needs to be rewritten.


6. So if you are considering the entire grassland of a county, it is not having any significant relationship with the drivers and in contrast if the total grassland is divided into low, medium and high grassland, it is having correlation with few drivers? Why is this happening. The drivers should have the same relationship even if you consider the entire grassland. What is the reason behind this result?

Additional comments

Additional comments can be found in the attached PDF.

Annotated reviews are not available for download in order to protect the identity of reviewers who chose to remain anonymous.

·

Basic reporting

No comment

Experimental design

No comment

Validity of the findings

No comment

Additional comments

"The article "Change Characteristics, Influencing Factors, and Suggestions of Grassland Degradation in Adjacent Areas" is a comprehensive and insightful examination of the current state of grassland degradation and the factors contributing to it. The authors have done an excellent job of presenting the information in a clear and concise manner, making it easy for readers to understand the complex subject matter.
The discussion of change characteristics is particularly noteworthy, as it provides a clear and detailed overview of the changes that have taken place in grasslands over time. This includes the physical changes such as soil degradation, loss of vegetation, and socio-economic factors that contribute to these changes, such as human activities, land use changes, and climate change.
The examination of influencing factors is also well-done, as it provides a thorough understanding of the various factors that contribute to grassland degradation. The authors have done an excellent job of connecting these factors to real-world examples, which helps to bring the information to life.
The suggestions for addressing grassland degradation are practical and well-supported by the evidence presented in the article. These suggestions include the need for better land-use planning and management, increased conservation efforts, and improved education and awareness programs.
Overall, this article is a must-read for anyone interested in understanding the current state of grassland degradation and the steps that can be taken to address it. The authors have done an excellent job of presenting the information in a clear and accessible manner, making it an important resource for researchers, policy-makers, and the general public alike."

Reviewer 3 ·

Basic reporting

This paper focused on the influenced factors of the grassland area and quality in adjacent areas of the Qinghai-Tibet Plateau, which had a certain novelty perspective. At the same time, the study found that the education level had a great impact on the grassland in different areas, which had a very interesting conclusion. Finally, the effective suggestions on grassland restoration would be suggested to the different local governments, which had great application value. Therefore, this study could prove to be a useful reference for the maintenance of grassland ecosystem. However, the language needs to be completely polished in the whole paper.

Experimental design

The part of “Data and Methods” was described with sufficient detail. It is quite suitable for the formation of some conclusions. However, it should be further added the calculation methods of the relevant parameters, such as the detailed calculation methods of ANPP, PNPP and HNPP, et al.

Validity of the findings

This paper discussed the changed characteristics o grassland and its influencing factors in 3 adjacent pastoral areas which in the transition zone of the northeast edge of the QTP and the western part of the Loess Plateau. The data was well analyzed, and good research results had been obtained. It's a comprehensive research report. However, there are still many details that need to be modified in the figures.
(1) The title grammars of Figure 4 and Figure 5 needs to be modified.
(2) The pictures in Figure 6 and Figure 12 are not clear, so it is recommended to upload the pictures again.
(3) It is recommended to reduce the colour saturation degree of Figure 14.

Additional comments

The author did much works for the paper, however, it was seem that more writing experiences and serious attitudes were needed for you. You should make changes to the English and grammar of this paper carefully.

---

## Round 0.2 · Minor Revisions

The manuscript is improved, but not yet ready for publication. Please check carefully the major comments suggested by reviewer #1. You may need to update the methods section(statistical trends, as suggested).

Reviewer 1 ·

Basic reporting

The introduction and discussion were well written. However, the language or sentence formations needs to be improved further. The figures and tables are of sufficient quality.

Experimental design

Even a single major comment suggested during the previous round were not addressed. I understand that implementing another method/model is time-consuming and increases the page-length. However, it would be appreciated to see at least one major change (may be simple Mann-Kendall and Sen's slope trends instead of linear trends) in the paper. Try to implement if possible or else check in your later manuscripts and report through another publication.

Validity of the findings

No comments

Additional comments

The minor comments and suggestions can be found in the attached PDF.

Annotated reviews are not available for download in order to protect the identity of reviewers who chose to remain anonymous.

Reviewer 3 ·

Basic reporting

no comments

Experimental design

It is OK.

Validity of the findings

The results are valid.

Additional comments

no comments

---

## Round 0.3 · Minor Revisions

Thanks for the manuscript update. The authors have addressed all of the reviewers' comments on the science part. However, there is a need to check English again. It demands minor revision, without a new reviewing round.

The language is not clear; there are some formatting and grammar mistakes. Please change active voice in English, not passive voice - for example, write “We show..” instead of “It was shown...”

Please check Abstract again, remove redundant abbreviations.

Here are some comments (line numbering is by ..Manuscript(tracked_changes).docx :

Line 14: “focused on a wide range,” - change wording in this phrase. Wide range of what?
“there was less researches..” - less number of publications? Please change this sentence.
Line 16: “This study investigated the changes..” - write as “We analyzed the changes...”. Not use ‘this study’
Line 19: “The results showed that...” - rephrase, change to “We found that..”, not use ‘the results’. It is not clear what are the results in the beginning of the text.
Line 22: “medium coverage grassland area..” - rephrase. Maybe area with ‘medium grass coverage’?
Than “was decreased” - change this wording
Line 24: ‘(ANPP)’ - remove this abbreviation since it is not used in the Abstract.
Line 26 ‘GDP’ - it is good give this abbreviation in full.
Line 35: “it need to improve..” - change wording
Line 37: add keyword “Qinghai–Tibet plateau” to the list of the keywords.
Remove keyword “suggestion”. May update keyword ‘education’, change it to ‘ecological education’
Line 48: “In China, the QTP area is approximately 2.5 million km2, which is nearly 25% of China's total area.” - not need repeat ‘China’ twice in one sentence. Change to “The area of QTP is approximately 2.5 million km2, which is nearly 25% of China's total area”.
Add a phrase about importance of QTP ecology and climate study after the phrase about large area size.
Line 52: “2019) , “ - remove space before ‘,’
Line 53: “2016) .” - remove space before ‘,’. Then in the text remove space before dot and comma sign. See line 65, line 71. It is just typo.
Line 125: “So, this study taken 3 adjacent areas (MQ, XH and LQ) in the east of the QTP as the research object.” - change the phrase, rearrange words. “We considered 3 adjacent areas..” or like that. Not use “as the research object”. Use plural form “objects”
Some symbols are not printed properly like ‘℃’ - for temperature (see line 145: “(5.07℃),” at least font changed in my computer.
Please check fonts are embedded in PDF file and all the symbols are correct in Word file.
Line 177: “The remote sensing data, which crucially for this study included the Normalized Difference Vegetation Index (NDVI).” - change this phrase, remove ‘which crucially for this study’. Add reference for MODIS and ARCGIS 10.2 software in the text phrases (web-link or literature).
Line 195: “...productivity was affected by climate, which is calculated from the Thornthwaite Memorial model.” - separate this sentence in two sentences. Add reference for Thornthwaite Memorial model...

Line 643: ‘Suggestions of grassland degradation..’
Change subtitle to ‘Suggestions of grassland restoration..”

Overall ,there are some language issue that needs text update, and then professional proofreading.
Please find professional agency or native English speaker to check.

As the editor I note not complete citation style (add access date for online resources, where appropriate), check the grammar.

Formulas (1-8) were removed. It is good to keep it in a short variant or move to some Supplementary file as a method part. Currently, there is no statistical modeling shown in the text. The problem of grassland area change and climate change is important. It is good to show novel more complex statistics, not just a correlation overview of previous studies.

---

## Round 0.4 · Minor Revisions

In the last decision letter, I pointed out some specific edits (which you have performed), but you did not do any other editing (despite the decision letter also telling you that a professional edit was required). However, we noticed that in the tracked changes doc, you have deleted (many) random words and then replaced them with the identical words, such that to a cursory look it appears that a lot of changes have been made.

In order to proceed, you must seek professional editing help and provide a certificate to show that this has been done.

---

## Round 0.5 · accepted · Accept

Thanks for the update. I see the detailed corrections on the text now. I believe the manuscript should be accepted in current form.